# A universal method for in situ control of stoichiometry and termination of epitaxial perovskite films

Bruce A. Davidson [1,2] ✉, Aleksandr Yu. Petrov[2], Fengmiao Li[1], Rebecca Pons [3], Pablo Sosa-Lizama[3], Hyungki Shin[1,4], Chong Liu [1], Pietro Parisse [2], Piero Torelli[2], Georg Cristiani[3], Y. Eren Suyolcu [3], Peter A. van Aken [3], Gennady Logvenov [3], Gideok Kim[3], Xiaoxing Xi [5], Eva Benckiser[3] ✉ & Ke Zou [1,4]

The perovskite $ABO_3$ structure serves as the foundation for diverse functional and quantum materials, yet its applications are hindered by challenges in control of film stoichiometry and the precise construction of interfaces, particularly compared to conventional semiconductors. While a layer-by-layer growth mode is frequently cited, we demonstrate that many transition-metal perovskite oxides self-assemble via an energetically favorable layer-inversion mechanism. This phenomenon can be strategically exploited to fine-tune stoichiometry and surface termination at any point during growth. Layer inversion produces consistent behavior in electron diffraction rocking curves and diffracted-beam intensity oscillations during alternating A- and B-site shuttered growth across various polar and nonpolar surfaces. We introduce a model that accurately interprets these oscillations, enabling an entirely in situ method for precise relative and absolute calibration of multielemental A- and B-site fluxes at the percent level. This approach is successfully applied to the growth of a single-phase high-entropy oxide film.

The $ABO_3$ perovskite structure, characterized by corner-sharing $BO_6$ octahedra and 12-fold coordinated A-site atoms at the cube corners (Fig. 1a), exhibits a wide array of electronic properties and functionally significant ferroic responses, which depend on the specific combinations of B- and A-site elements—typically transition metals and alkaline or rare-earth metals, respectively[1]. Additionally, octahedral distortions away from the ideal cubic configuration can further influence these properties[2]. The stable bonding network of perovskites allows for considerable off-stoichiometry in cation ratios (A:B ≠ 1) and oxygen content ($O_{3±δ}$) while maintaining long-range perovskite symmetry, as demonstrated in materials such as $SrTiO_3$ (STO) and $EuTiO_3$[3–5], manganites[6] and nickelates[7]. Moreover, significant oxygen excess or deficiency can be stable[8,9] and ordered, as seen in the "Brownmillerite"

phase $SrCoO_{2.5}$[10,11] and the "infinite-layer" $ABO_2$ phases found in cuprates and nickelates[12–14]. The ability of these materials to accommodate large compositional deviations can obscure the intrinsic properties of stoichiometric layers and interfaces, as well as the role of defects, highlighting the critical need for enhanced growth and characterization techniques for these multi-elemental compounds.

In molecular-beam epitaxy (MBE), simultaneous deposition of all cations ("codeposition") relies on precise control of elemental A- and B-site fluxes by setting individual cell temperatures and monitoring with a quartz-crystal microbalance (QCM)[15–17]. However, QCM inaccuracies, particularly for volatile elements[18] and flux variations in oxidizing environments[19], have led to extensive investigations into alternative calibration methods for many common elements using their binary

[1]Quantum Matter Institute, University of British Columbia, Vancouver, Canada. [2]CNR-IOM, Istituto Officina dei Materiali, Area Science Park, Trieste, Italy. [3]Max Planck Institute for Solid State Research, Stuttgart, Germany. [4]Department of Physics and Astronomy, University of British Columbia, Vancouver, Canada. [5]Department of Physics, Temple University, Philadelphia, PA, USA. ✉e-mail: bruce.davidson@ubc.ca; E.Benckiser@fkf.mpg.de

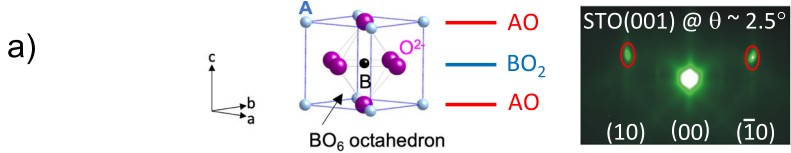

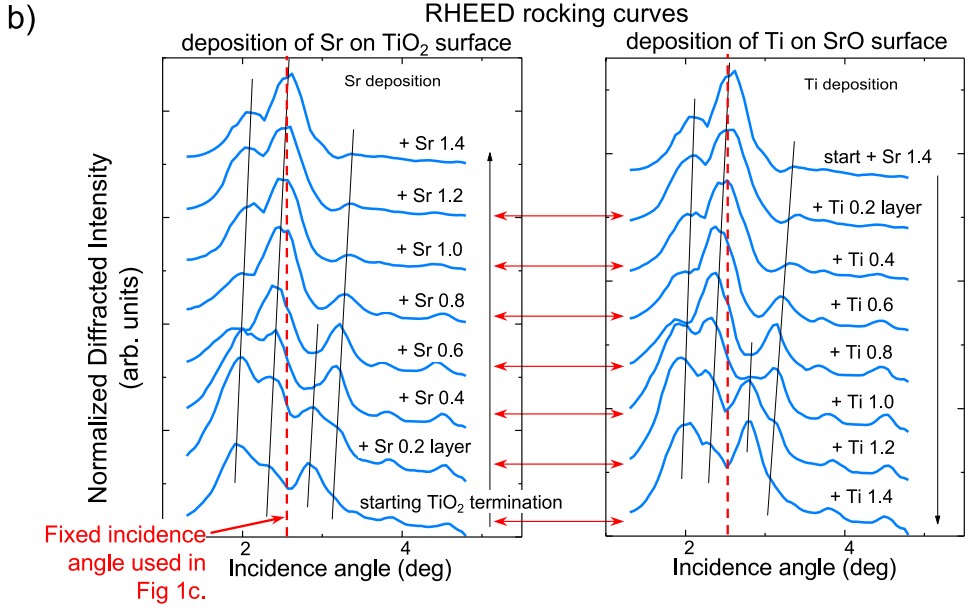

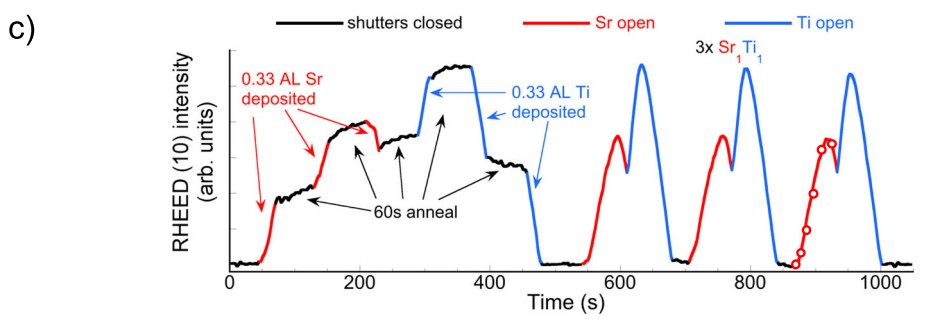

**Fig. 1 | Dynamic layer rearrangement revealed in RHEED rocking curves during shuttered deposition.** The diffracted-beam rocking curve shape depends only on net Sr coverage on the $TiO_2$ termination and is stable during anneals. **a** RHEED on perovskites. Layered $ABO_3$ crystal structure along (001) growth direction, and RHEED image of $TiO_2$-terminated $SrTiO_3$(001) surface with beam along (100) azimuth. Diffracted reflections are encircled by red ellipses inside which the (average) intensity is monitored; specular (0,0) reflection is also indicated. **b** RHEED diffracted intensity rocking curves. Left panel: rocking curves during Sr deposition starting on $TiO_2$ termination. Curves are measured during growth interruptions after Sr deposition in 0.2 monolayer increments, repeated to a total of 1.4 Sr layers; intensities are normalized to the maximum of each scan. Right panel: return to $TiO_2$ termination by Ti deposition in 0.2 monolayer increments. Note the similarity of the rocking curves for the same net Sr coverage (e.g., compare "Sr +0.6" and "Sr +1.4 followed by Ti +0.8"). Gray lines are guides to the eye for the peak positions, and curves are shifted for clarity. **c** RHEED diffracted intensity oscillations during shuttered STO growth of $Sr_1/Ti_1$ cycles (Sr in red, Ti in blue) at ~2.5° beam incidence (vertical dashed line in (**b**)). The first cycle is broken into 3 segments of Sr (red) or Ti (blue) deposition, separated by anneals (black), followed by 3 complete $Sr_1/Ti_1$ cycles. The starting surface has ~0.5 SrO atomic layer on $TiO_2$ termination. On the last cycle, open circles (860–920 s) are intensities taken from the rocking curves at equivalent Sr coverage.

oxides[5]. In contrast, single-target pulsed laser deposition (PLD)[20] and sputter deposition[21] solve the composition problem by identifying conditions that promote stoichiometric transfer to the substrate[22–27]. Notably, "hybrid" MBE has shown growth regimes in which stoichiometric, self-regulating growth of certain perovskites can be achieved[28–30]. Across all these techniques, in situ reflection high-energy electron diffraction (RHEED)[31–33] has enhanced real-time characterization of the growing surface, significantly improving deposition control whether by MBE[34–37], PLD[38] or sputtering[39]. At 10 keV energies and a few degrees of incidence, the RHEED beam typically probes several unit cells, and extracting detailed structural information such as surface atomic positions from RHEED data is complicated[31,32]. As a result, although RHEED is ubiquitous in advanced oxide growth systems, it is

primarily utilized to assess substrate or film surface quality and to calibrate growth rates during codeposition by analyzing the period of specular intensity oscillations, which is often cited as evidence of layer-by-layer growth by analogy with original semiconductor studies[40,41]. Given these limitations, significant advances in understanding perovskite film growth processes have largely relied on in situ synchrotron surface X-ray diffraction, initially in studies of codeposition via PLD[42–46].

Unlike codeposition, the layered structure of perovskite unit cells suggests the possibility of sequential deposition of A- and B-site cations in an "alternating-shutter" approach. This method was pioneered by MBE[47–49] and later adapted for PLD using binary oxide targets[50–52]. Shuttered growth offers advantages over codeposition by allowing, in principle, control over the stacking order of layers, in particular at

interfaces, thereby promoting synthesis of phase-pure layered materials and precise construction of superlattices[53–55]. However, in situ synchrotron-based crystal truncation rod (CTR) analysis during shuttered MBE growth of $SrTiO_3$ (STO) films has shown that the deposited layers do not always maintain the intended order dictated by the shutter sequence, instead revealing an A-site/B-site layer inversion mechanism[56,57]. In this process, a $SrO$-$SrO$-$TiO_2$ sequence deposited on a $TiO_2$ surface reassembles dynamically to $SrO$-$TiO_2$-$SrO$. Layer inversion is thermodynamically driven and occurs for combinations of B- and A-site cations for which the $AO$-$BO_2$-$AO$ stacking pattern is energetically stable[56]. Layer inversion during the growth of STO has been confirmed through electron microscopy[58], as well as by in situ CTR analysis during $LaTiO_3$[57] and nickelate growth[59,60]. These studies demonstrate that shuttered growth allows for advanced control over stoichiometry, layer stacking, and surface termination when combined with synchrotron surface diffraction measurements during deposition.

In this study, we present a phenomenological model for interpreting RHEED intensity oscillations during shuttered perovskite growth that provides an alternative to synchrotron-based techniques for precise stoichiometry and surface termination control. Derived from RHEED rocking curves, the model applies to many nonpolar and polar phases, including mixed-valence titanates, manganites, ferrites, and nickelates, in which layer inversion occurs. The rocking curves alone allow precise determination of the surface termination, which can subsequently be adjusted to pure $BO_2$, $AO$, or any mixture prior to heterointerface growth. From the model, we develop a universal method for interpreting RHEED oscillations during shuttered growth that enables precise control of relative stoichiometry (A:B = 1) per cycle. The method exploits dynamic layer rearrangement, intentionally inducing layer inversion by initiating the shuttered growth cycle with excess A-site on the $BO_2$-terminated surface. By combining shuttered growth with codeposition, we establish a fully in situ method for accurate calibration of both relative and absolute stoichiometry ($A_{1.00}B_{1.00}$) using only RHEED. This approach is successfully applied to grow a precisely stoichiometric high-entropy oxide film ($(Sr_{0.25}Eu_{0.25}La_{0.25}Nd_{0.25})TiO_3$), underscoring its potential to advance growth control for functional perovskite films and heterostructures beyond current methods.

## Results

In the typical RHEED geometry (Supplementary Fig. 1), a rocking curve measures the intensity (specular or diffracted) as a function of incidence angle, and its shape is highly sensitive to crystal structure, surface reconstructions and elemental distribution, especially in layered structures[31,32]. Diffracted-beam rocking curves taken during interruptions of a shuttered $Sr_{1.4}/Ti_{1.4}$ growth cycle (Fig. 1b) show a series of peaks that evolve systematically as the termination changes, ultimately reverting to its initial shape and intensity as the termination returns to the starting $TiO_2$. Here, $Sr_x/Ti_y$ denotes a deposition cycle with $x$ atomic layers of $SrO$ followed by $y$ layers of $TiO_2$, and only diffracted intensities are analyzed as they are less sensitive to step-edge density compared to the specular intensity[47–49,52].

Dynamic rearrangement of layers is apparent from the evolving shapes of rocking curves during incremental addition of $AO$ or $BO_2$ layers. A comparison of rocking curves at intervals in the deposition cycle (Fig. 1b) indicates that their shape depends on the surface termination—whether pure or mixed—independent of whether the same termination is achieved through deposition of Sr on $TiO_2$ or Ti on $SrO$ surfaces. Similar to findings from X-ray CTR studies on titanates[56,57], the distinct RHEED rocking curve shapes at specific stages in the shuttered cycle act as a fingerprint to identify the mixed surface composition, reflecting the partial $SrO$ coverage on a complete $TiO_2$ layer. These rocking curves also suggest that the surface state remains stable during pauses in deposition, with nearly all layer interdiffusion completed during a single unit-cell cycle (~100 s), within the sensitivity of RHEED and consistent with the CTR studies[57].

The rocking curves show a complicated series of broad peaks, not reproduced by calculations using a simple kinematic scattering model[48], that shift reproducibly in position and whose intensities increase and decrease dramatically during the cycle. Similar conclusions are seen for rocking curves during shuttered growth of the polar perovskite $LaFeO_3$ (Supplementary Fig. 2). By quantifying the mixed termination through these rocking curve shapes, an approximate accuracy of ±0.1 A-site layers is achievable, offering an approach to estimate the termination in static conditions, i.e., without deposition. As shown later, greater precision can be achieved by analyzing the RHEED intensity oscillations during continuous shuttered cycles, allowing for fine control over surface termination and stoichiometry in complex oxide film growth.

Due to the stability of the surface chemical-structural states observed in the RHEED data, a full set of rocking curves taken with small deposition increments can be used to reconstruct the intensity oscillations during continuous shuttered cycles at any fixed incidence angle. Choosing an incidence angle at which the diffracted intensity shows its largest amplitude variations, four $Sr_1/Ti_1$ cycles are deposited (Fig. 1c). The nearly-identical Sr/Ti (red/blue) curves with and without interruptions again confirm that a stable surface state is reached within the cycle time. Moreover, after complete cycles in which equal amounts of Sr and Ti are deposited, the RHEED intensity returns to its starting intensity, implying a recovery of the starting termination. Under optimized conditions, oscillations remain consistent in amplitude and shape across hundreds of cycles, enabling precise and long-term monitoring of growth parameters through RHEED.

The model in Fig. 2 explains the interpretation of RHEED diffracted intensity oscillations during $Sr_x/Ti_x$ cycles with different surface terminations, providing a phenomenological framework for controlled (001)-oriented growth of various $ABO_3$ perovskites. At the chosen incidence angle (~2.5°), the diffracted intensity shows its largest amplitude variations, and the oscillation behavior is robust against small misalignment (see Discussion). Experimentally, this angle also corresponds to the maximum intensity of the specular rocking curve for $TiO_2$ termination, and where the specular reflection crosses the primary Kikuchi lines (Fig. 1a). This geometry leverages Kikuchi lines, whose position is independent of angle[32], as an internal reference to set the incidence angle without additional calibration. That the strongest diffracted intensity oscillations and maximum specular intensity occur at the same incidence angle is a consequence of dynamical scattering of electrons that produce the RHEED pattern under the geometric constraints of the layered perovskite (001) structure. Evidence suggesting this incidence corresponds to a RHEED surface resonance is discussed later.

The model in Fig. 2a is derived from the experimental data in Fig. 2b, which shows diffracted intensity oscillations for different $Sr_x/Ti_x$ cycles ($Sr_1/Ti_1$, $Sr_{1.5}/Ti_{1.5}$, $Sr_2/Ti_2$ and $Sr_{2.5}/Ti_{2.5}$). As $x$ increases above ~1.3, the oscillations develop a distinct "double-peak" shape, with one inflection point (maximum) per element. Consequently, if the $SrO$ coverage on $TiO_2$ exceeds 1.3 layers at any point during a cycle, the double-peak shape will be present. The model can describe oscillations for different $x$ values, regardless of the starting surface termination, as follows.

In the model, the reference curve is the "deep" double-peak oscillation seen during a $Sr_3/Ti_3$ deposition cycle starting on $TiO_2$ termination (red/blue dashed lines in Fig. 2a). When a $Sr_1/Ti_1$ or $Sr_2/Ti_2$ cycle is deposited, the Sr and Ti shutters will be open the corresponding fraction of the $Sr_3$ and $Ti_3$ shutter times, and the observed RHEED intensity will trace only part of the $Sr_3$ and $Ti_3$ curves. The resulting oscillation for a $Sr_1/Ti_1$ cycle is triangular (upward on Sr/downward on Ti) while $Sr_{1.5}/Ti_{1.5}$ or $Sr_2/Ti_2$ cycles show a shallow or deep double-peak shape. In the generic model of Fig. 2a, the maximum intensities at the A- and B-site inflection points are represented as equal, while in practice these intensities can be different (Fig. 2b, d for

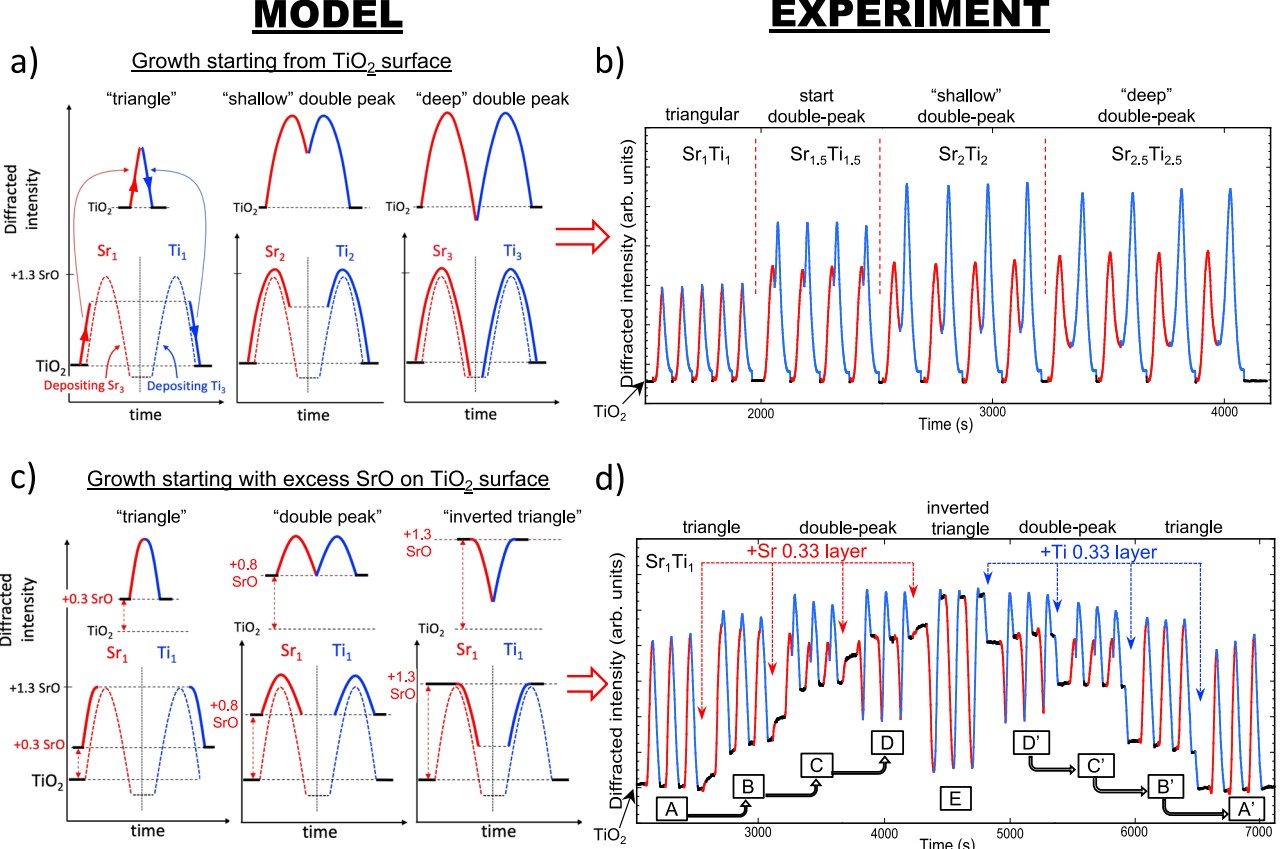

**Fig. 2 | Universal model for RHEED diffracted-intensity oscillations during alternating-shutter growth. a** Model: intensity oscillations starting from a $TiO_2$-terminated surface. 3 panels showing the oscillation shape during one deposition cycle of $Sr_1/Ti_1$, $Sr_2/Ti_2$ and $Sr_3/Ti_3$. The oscillation of a full $Sr_3/Ti_3$ cycle (dashed red/blue lines) serves as the reference curve for $Sr_x/Ti_x$ cycles ($x < 3$). A cycle of $Sr_1/Ti_1$ or $Sr_2/Ti_2$ will trace the corresponding fraction (solid lines) of the reference $Sr_3/Ti_3$ curves. During Sr deposition, an inflection point (maximum) occurs at ~1.3 Sr layers on the $TiO_2$ surface (**b**). During a $Sr_1/Ti_1$ cycle, the inflection point is not reached and the oscillation traces a triangular shape (up on Sr/down on Ti); a $Sr_2/Ti_2$ cycle traces a "shallow double-peak" shape; and a $Sr_3/Ti_3$ cycle traces the full "deep double-peak" shape. At the end of each cycle, the intensity returns to its starting value corresponding to $TiO_2$ termination. Colors: red = Sr shutter open; blue = Ti shutter open; black = all shutters closed. Fluxes are assumed constant. **b** Experiment: diffracted intensity during $Sr_x/Ti_x$ cycles on $TiO_2$ termination. Diffracted intensity during a sequence of $Sr_x/Ti_x$ cycles, with $x = 1$, 1.5, 2 and 2.5, showing triangular and double-peak oscillations. **c** Model: intensity oscillations starting from mixed termination. 3 panels showing oscillation shapes during a $Sr_1/Ti_1$ cycle, starting from different partial SrO coverages (0.3, 0.8 and 1.3 layers) on a $TiO_2$ surface. The

observed intensity (solid lines) follows the universal curves for a $Sr_3/Ti_3$ cycle (dashed lines), starting at the point on the curve corresponding to the partial Sr coverage, and tracing an arc for one-third of the $Sr_3$ ($Ti_3$) shutter time. Consequently, a $Sr_1/Ti_1$ cycle starting from 0.3 SrO layers/$TiO_2$ shows an upward triangular shape (up on Sr/down on Ti); a cycle starting from 0.8 Sr layers/$TiO_2$ passes the inflection point and shows a symmetric "double-peak" shape; and a cycle starting from 1.3 Sr layers/$TiO_2$ shows an downward triangular shape (down on Sr/up on Ti). **d** Experiment: intensity oscillations for different mixed terminations. Diffracted intensities during 3 cycles of $Sr_1/Ti_1$ for different partial SrO coverage on $TiO_2$: starting from [A] $TiO_2$ termination, [B] 0.33 layer SrO, [C] 0.66 SrO, [D] 1.0 SrO and [E] 1.33 SrO. The original $TiO_2$ termination is recovered by depositing Ti in increments of 0.33 layers [D′]→[C′]→[B′]→[A′]. During the deposition of each partial 0.33 SrO layer, the intensity shows a step-like increase; and during each partial 0.33 $TiO_2$ layer, the intensity shows a step-like decrease. Note the similar shape of oscillations obtained on surfaces with similar net Sr coverage, e.g., [A] and [A′], [B] and [B′], etc. The oscillation shapes agree with those predicted by the model in (**c**) for similar net Sr coverage, i.e., triangular (starting Sr = 0 and 0.33), shallow double-peak (Sr = 0.6), deep double-peak (Sr = 1.0), and downward triangle (Sr = 1.33).

STO). This does not affect the interpretation of the oscillations when applying the model.

The RHEED oscillation shape for a $Sr_1/Ti_1$ cycle starting from any mixed termination can be predicted by the model (Fig. 2c): a given starting SrO coverage only changes the starting point on the reference curve from which the arcs begin. Depending on the starting coverage, the oscillation shape can be triangular up, "double-peak", or triangular down. Experimentally (Fig. 2d), the different shapes predicted by the model can be observed by controllably changing the termination before starting $Sr_1/Ti_1$ cycles: triangular up ([A] and [B]), shallow ([C]) or deep double-peak ([D]) or triangular down ([E]). These shapes are reproduced in reverse by incremental $TiO_2$ deposition ([D′]→[C′]→[B′]→[A′]). Overall, the oscillations in Fig. 2d agree well with those predicted by the model. Complete layer inversion during the entire sequence of Fig. 2d is confirmed by equal starting and ending

intensities (before [A] and after [A′]) corresponding to $TiO_2$ termination, and by the X-ray diffraction results presented below.

The model shows that upward or downward triangular shapes for $A_1/B_1$ cycles generally indicate a starting surface near the B-site or A-site termination, respectively, and "frequency-doubled" oscillations reveal a starting surface with roughly half an A-site layer on the $BO_2$ surface. The precise shape depends on the A-site coverage at inflection, which for STO corresponds to 1.3 SrO/$TiO_2$ at this incidence angle. Once the coverage at inflection is known, the model allows identification of the starting surface termination from the oscillation shape if the fluxes are known.

Interpretation of shuttered oscillations from the model (Fig. 2) forms the basis of a universal calibration procedure (Fig. 3) that allows determination of both relative stoichiometry and absolute layer doses for multiple A- and B-site elemental fluxes from parameters measured entirely in situ. This contrasts with ex situ flux calibration of

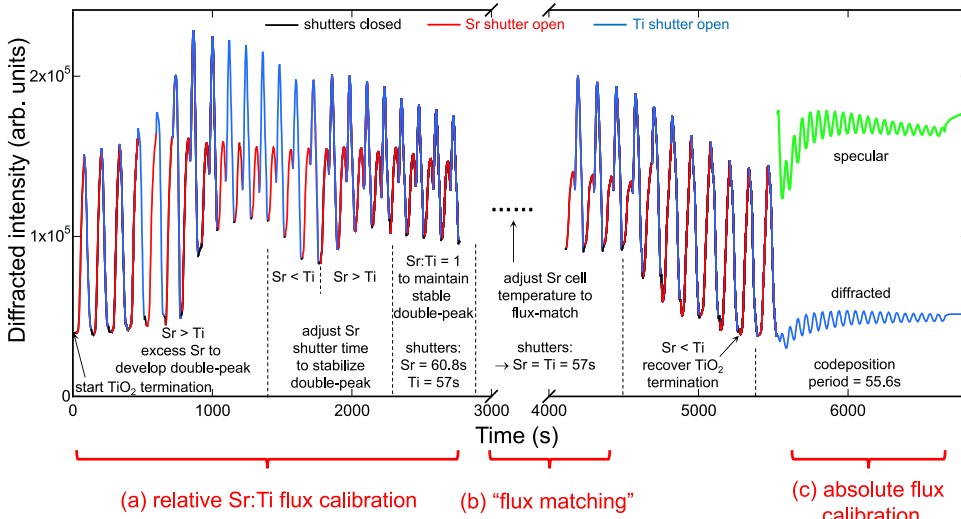

**Fig. 3 | General method for relative and absolute calibration of fluxes using diffracted intensity oscillations. a** Determining the relative stoichiometry (Sr:Ti = 1) by shuttered growth. RHEED diffracted intensity oscillations (red/blue) starting from the TiO$_2$-terminated surface. Cycles with excess Sr ("Sr > Ti", nominally Sr$_{1.10}$/Ti$_1$) are deposited until sufficient partial A-site coverage has accumulated on the growing surface for the"double-peak" shape to appear (~800 s). Then the Sr shutter time is adjusted until the "double-peak" shape becomes stable and repeatable: if the Sr dose per cycle is larger than the Ti dose, the dip in the double-peak will become progressively deeper (800–1400 s and 1700–2300 s); if the Sr dose is smaller than the Ti dose then the dip in the double-peak will become progressively more shallow (1400–1700 s). Once the doses are equal (ratio Sr:Ti = 1), then the same double-peak shape will repeat for multiple cycles (2400–2800 s and 4200–4500 s). This determines the A- and B-site shutter times for stoichiometric ABO$_3$ deposition. Note that, at this point, the dose $x$ in Sr$_x$/Ti$_x$ cycles is usually different than 1. **b** Matching Sr and Ti fluxes. Following (**a**), the Sr cell temperature is adjusted while depositing, maintaining the same "double-peak" shape, until the Sr and Ti shutter times per cycle are equal (~4200 s; here the Sr cell was increased by 1.9 °C). Equal shutter times define "flux-matched" conditions for SrTiO$_3$, i.e., $F_{Sr} = F_{Ti}$, where $F_{Sr}$ and $F_{Ti}$ are the Sr and Ti fluxes in atoms/(cm$^2$·s). **c** Determining the absolute flux calibration by codeposition (each shuttered cycle deposits precisely one unit cell of SrTiO$_3$). After setting the surface to TiO$_2$ termination by depositing five cycles of Sr$_{0.92}$/Ti$_1$, both shutters are opened simultaneously (5500 s). The oscillation period is fit, yielding the shutter time for deposition of one complete unit cell of SrTiO$_3$. This shutter time is used for both Sr and Ti in subsequent shuttered growth. The absolute flux calibration, e.g., for homoepitaxial growth on STO, is then $F_{Sr} = F_{Ti} = 6.6 \times 10^{14}$ atoms/cm$^2$ divided by the shutter time. From the Sr and Ti calibrations, the absolute calibration of other fluxes, e.g., La, can be found by applying the first step of this procedure to the growth of Sr$_{1-x}$La$_x$TiO$_3$ for one or more values of $x$ (see Supplementary Note 1).

Ruddlesden-Popper phases employing X-ray diffraction and reflectivity on calibration films prior to growth[54,55]. The procedure presented here offers similar accuracy (~1% error in absolute fluxes) and has the additional advantage that any drift in fluxes can be seen and corrected in real time during growth.

The universal calibration method consists of three steps, shown in Fig. 3 during STO growth. First, a Sr-rich surface is intentionally prepared, then cycles of Sr$_x$/Ti$_x$ ($x$ - 1) are deposited, and the shutter times are determined that keep the double-peak shape constant; this indicates growth of stoichiometric STO. Second, one cell temperature is adjusted until the shutter times are equal, again keeping a constant double-peak shape; this ensures that Sr and Ti fluxes are equal. Third, Sr and Ti are codeposited; the period of the intensity oscillations then gives the absolute flux calibration. These steps form the basis of the method. The calibration of one perovskite phase (e.g., SrTiO$_3$) allows calibration of other fluxes to create mixed-valence phases via A- or B-site cation substitution (Supplementary Note 1).

In the first step, the Sr and Ti shutter times are adjusted until stable double-peak oscillations are established; this gives the relative flux calibration and ensures equal doses (atoms/cm$^2$) of Sr and Ti per cycle. Flux-matching Sr and Ti in the second step, by adjusting one cell temperature, is necessary to perform codeposition in the third step. The oscillation period during codeposition determines the shutter time required for deposition of one complete unit cell (A$_{1.00}$B$_{1.00}$). This last step provides the absolute flux calibration of each element (in atoms/cm$^2$·s). Stable codeposition oscillations after flux-matching confirm that the shutter method indeed results in correct stoichiometry[61], as discussed below.

The calibration procedure requires adjusting A- and B-site shutter times and cell temperatures to maintain a constant "double-peak" shape in the diffracted intensity oscillations. Since this shape is extremely sensitive to flux variations (Supplementary Note 1), any drift in fluxes can be detected by tracking the dip intensity between peaks and making slight adjustments to the shutter times to keep it constant. After calibration and under optimized growth conditions, the RHEED diffracted intensity represents the integrated dose, i.e., with unit-cell doses, reaching the same dip and final intensities in successive cycles implies that identical quantities of A- and B-site atoms/cm$^2$ are deposited per cycle, regardless of flux variations that may occur during the cycle. We estimate careful adjustment of shutter times can maintain stoichiometry within 0.5% of A:B = 1 per cycle; small variations in dip intensity between cycles only indicate slight differences in partial A-site coverage on the BO$_2$ surface after each cycle.

Ex situ characterization confirms that the shutter method produces stoichiometric, high-quality films whose long-range order is limited primarily by substrate quality. High-resolution X-ray diffraction (XRD) 2θ-ω scans of STO films grown on STO(001) substrates using Sr$_1$/Ti$_1$ cycles and starting each cycle with ~0.5 layer SrO/TiO$_2$ (e.g., Figs. 1c or 3) show indistinguishable film and substrate (00*l*) peaks and identical rocking curve FWHM (Supplementary Fig. 5), no observable secondary phases, and no thickness oscillations in X-ray reflectivity (XRR) scans[62]. Furthermore, deposition cycles up to Sr$_{2.5}$/Ti$_{2.5}$ result in high-quality STO films (Fig. 4 and Supplementary Fig. 6), giving further evidence that layer inversion is active and complete during a deposition cycle even when Ti diffusion occurs through more than one SrO layer[63].

Cross-sectional scanning transmission electron microscopy (STEM) analysis (Fig. 4) further confirms the high epitaxial quality. High-angle annular dark-field (HAADF) images (Fig. 4a) and electron energy loss spectroscopy (EELS) elemental maps (Fig. 4b, c) show a defect-free film structure and a homogeneous elemental distribution.

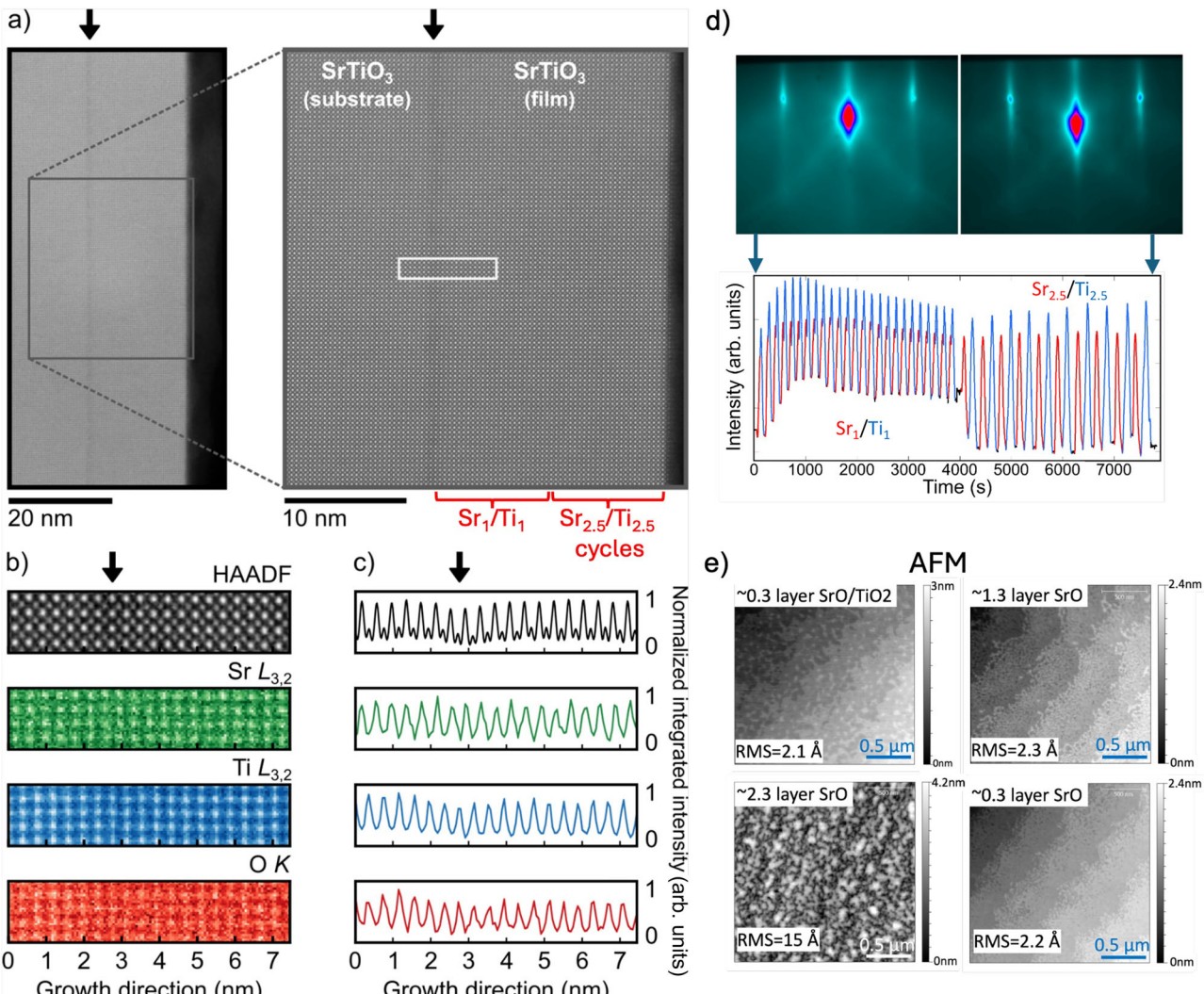

**Fig. 4 | STEM, RHEED and AFM on shutter-grown STO//STO(001). a** STEM-HAADF images of ~20 nm SrTiO₃ thin film showing high epitaxial quality and a slightly darker contrast at the substrate-film interface region. Low magnification (left) highlights uniformity of the film, and high magnification (right) presents an atomically resolved image of the defect-free film. Deposition was started with 25 cycles of $Sr_1/Ti_1$ followed by 10 cycles of $Sr_{2.5}/Ti_{2.5}$. **b** HAADF image and EELS elemental maps across the interface region (white rectangle in (**a**)). In the HAADF image, the cation columns give rise to bright contrast, corresponding to the Sr (green) and Ti (blue) elemental maps below. The O map (orange) shows a delocalized intensity. **c** Intensity profiles integrated from the maps along the growth direction; a dip in the HAADF intensity at the interface is visible, coinciding with a similar feature in the Ti (blue) and O (orange) profiles, while no variations are observed in the Sr profile. The dip in the HAADF intensity at the interfacial region is consistent with RHEED images in (**d**). **d** RHEED images of the disordered STO substrate surface and improved film surface after growth. RHEED oscillations show $Sr_1/Ti_1$ and $Sr_{2.5}/Ti_{2.5}$ cycles used during the first and second halves of growth. **e** AFM images of the STO surface after interruption at different points in the shutter cycle. Terrace structure and root-mean square (RMS) roughness (~2 Å) for all scans is similar to the starting substrate, except for higher roughness (~15 Å) deep in the dip of the double-peak cycle (~2.3 layers SrO/TiO₂). These images suggest that the decrease in the diffracted intensity after the Sr inflection point (~1.3 layers SrO/TiO₂) in the $Sr_x/Ti_x$ cycle may be due, at least in part, to increased surface roughness [52].

The interface region shows a slight decrease in the HAADF intensity that is seen in the atomically resolved EELS elemental maps, corroborating the weaker RHEED features of the starting substrate (Fig. 4d). The disordered substrate surface, likely due to the surface preparation of these commercial substrates, is recovered after five unit cells and the remaining film is uniformly stoichiometric and highly crystalline (Fig. 4a), further evidence that growth with cycles of $Sr_1/Ti_1$ or $Sr_{2.5}/Ti_{2.5}$ is indistinguishable.

Taken together, these results give clear evidence that dynamic layer rearrangement prevents retention of SrO-SrO bilayers, which are present each cycle and are stable in rock-salt SrO or Ruddlesden-Popper phases. Atomic force microscopy (AFM) images (Fig. 4e) show atomically flat, terraced surfaces for Sr coverages below ~1.3 layers, with roughness comparable to the starting TiO₂-terminated surface; rougher surfaces are seen in the "dip" region (between Sr and Ti inflection points). XRR on heteroepitaxial films confirms absolute calibration accuracy to ~1% of the film thickness (Supplementary Note 1 and Supplementary Fig. 4). Rutherford backscattering spectrometry (Supplementary Fig. 7) confirms global stoichiometry and areal densities.

RHEED "double-peak" oscillations appear during MBE growth of a wide range of ABO₃ perovskites, including mixed-valent titanates, manganites, ferrites, nickelates, aluminates and zirconates (Fig. 5, Supplementary Figs. 8–12). This allows the calibration method in Fig. 3 to be applied across the full phase diagram $0 \le x \le 1$ of $La_{1-x}Sr_xTiO_3$, $La_{1-x}Eu_xTiO_3$[64], $La_{1-x}Sr_xMnO_3$[65], $Nd_{1-x}Sr_xTiO_3$ and $La_{1-x}Sr_xFeO_3$, enabling precise control of doping $x$. It also extends to isovalent phases of titanates such as $Eu_{1-x}Sr_xTiO_3$, $Ba_{1-x}Sr_xTiO_3$, $Ca_{1-x}Sr_xTiO_3$ and $Ba_xCa_ySr_zTiO_3$ ($x+y+z=1$, $0 \le x,y,z \le 1$) and zirconates such as

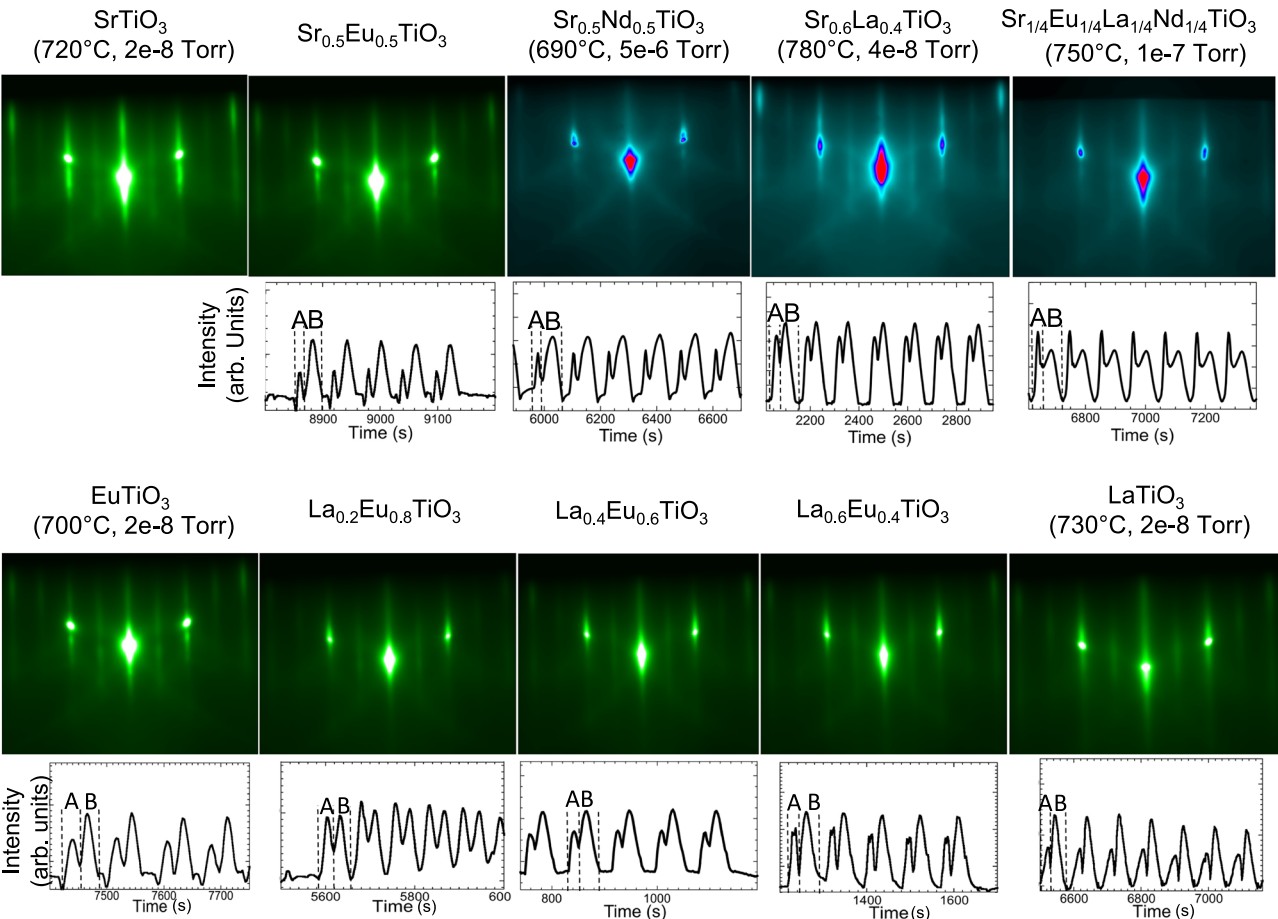

**Fig. 5 | RHEED images and diffracted intensity oscillations during shutter growth.** RHEED images taken at the end of an A/B cycle, and double-peak oscillations used for calibrating relative stoichiometry during shutter growth, for a range of titanate perovskites. The data are taken from the growth of films across the phase diagrams of solid solutions of SrTiO3, EuTiO3, LaTiO3 and NdTiO3, including the high-entropy oxide Sr0.25Eu0.25La0.25Nd0.25TiO3. All films grown on STO(001).

Ba$_{1-x}$Ca$_x$ZrO$_3$. Note that LaAlO$_3$ and CaZrO$_3$ films grown by PLD in Supplementary Fig. 12 demonstrate that layer inversion also occurs in PLD growth, and the model in Fig. 2 is equally applicable to films grown by sequential ablation from binary oxide targets. We further note that manganite tunnel junctions grown by MBE using this method have shown amongst the highest reported tunneling magnetoresistance values for perovskite-based devices to date[66]. This method also provides the basis for the first precisely-controlled deposition by MBE of multi-elemental high-entropy oxide (HEO) phases like (Sr$_{0.25}$Eu$_{0.25}$La$_{0.25}$Nd$_{0.25}$)TiO$_3$ (Fig. 5, Supplementary Note 1, Supplementary Figs. 3, 4). High-entropy titanate phases display promise as novel thermoelectric materials[67,68]. In practice, calibration for multiple materials and growth optimization over a wide range of oxygen pressures and substrate temperatures can be completed on the same substrate within a few days. Daily recalibration takes a few hours or less. Since this method determines absolute fluxes for all elements, judicious matching of fluxes allows the growth of other nonperovskite phases (e.g., spinel, pyrochlore, Aurvillius) by codeposition.

## Discussion
The ubiquitous presence of RHEED double-peak oscillations in shuttered growth of both nonpolar A$^{2+}$B$^{4+}$O$_3$ (e.g., SrTiO$_3$, EuTiO$_3$, SrMnO$_3$, SrFeO$_3$, BaZrO$_3$) and polar A$^{3+}$B$^{3+}$O$_3$ (e.g., LaTiO$_3$, NdTiO$_3$, LaFeO$_3$, LaMnO$_3$, LaNiO$_3$, LaAlO$_3$) phases is perhaps unexpected, because it is likely that instabilities at a polar surface[69] may modify the microscopic growth mechanisms (oxidation states, diffusion kinetics, surface reconstructions, etc.) in a complicated way. The applicability of the model (Fig. 2) requires both layer inversion and the use of the specific RHEED diffraction condition. Layer inversion during shuttered perovskite growth has been predicted for a number of B-site elements and Sr or La as A-site elements[56] and was excluded for La-based manganites and ferrites. However, the data presented in Supplementary Figs. 9 and 10 show these families also exhibit layer inversion. In general, layer inversion may be tied to the stability of Ruddlesden-Popper phases (or more precisely, double-rock-salt AO layers) within a given perovskite family. While many 3 d and 4 d transition metals form Ruddlesden-Popper phases, not all have been reported. Further spectroscopic studies during growth may help clarify the microscopic mechanisms underlying layer inversion and the conditions for its occurrence.

Second, the diffraction condition utilized in Fig. 2 is selected to maximize the amplitude of diffracted intensity oscillations during A$_x$/B$_x$ cycles, enhancing sensitivity to elements in the topmost layer. The strong sensitivity suggests a surface resonance at this incidence, arising from dynamical scattering within the layered crystal structure—an effect similar to those observed in RHEED studies on semiconductors[32]. As supporting evidence, the double-peak shape remains with comparable intensity when the azimuthal angle is rotated a few degrees away from (100) toward the "one-beam" condition. In this condition, the intensity is primarily influenced by layer spacing along the surface-normal direction and the average layer density[32]. During (001) perovskite growth, the alternating AO and BO$_2$ planes create an average density contrast that will depend on the masses of the A- and B-site elements. This can explain why intensity increases

when heavier A-site atoms are deposited on a lighter B-site termination and decreases vice versa, forming the "upward triangle" shape at this incidence angle seen in Fig. 2.

In addition, a larger A-site scattering factor is a common feature of the phases studied here. This is related to the structural stability criterion for perovskites, which generally requires a larger ionic radius for A-site than B-site atoms. Stability is quantified by the Goldschmidt tolerance factor $t = (r_A + r_O)/\sqrt{2}(r_B + r_O)$, in which $r_A$, $r_B$ and $r_O$ are the ionic radii of the A-site, B-site and oxygen atoms, respectively[70,71]; perovskites are typically stable in the range $0.75 < t < 1.05$[72]. Larger ionic radii and scattering factors of alkaline- and rare-earth elements compared to 3d and 4d transition elements in most perovskites suggest that the "upward triangle" behavior during an $A_1/B_1$ cycle starting from $BO_2$ termination will be a nearly-universal feature of shuttered growth. However, further studies are needed to confirm this experimentally.

Lastly, our results can be compared to previous studies of shuttered RHEED oscillations along the pseudocubic (110) azimuth to control film stoichiometry[47,48] in contrast to the (100) azimuth used here[49,52,65]. Along (110), different oscillation shapes (triangular, frequency-doubled, inverted triangular) were attributed to different incidence angles[48]. However, along (100) all these shapes can be reproduced at a single incidence angle by adjusting the A-site coverage on the $BO_2$ surface at the start of each cycle (Fig. 2). It is well-known that mixed terminations inherited from the substrate or accumulated by off-stoichiometric growth are common in complex oxide film growth by all techniques. Furthermore, our results suggest that alignment along (100) may offer advantages for stoichiometry and termination control. Supplementary Figs. 13 and 14 compare RHEED double-peak oscillations as a function of incidence angle along the (100) and (110) azimuths. Along (100), the upward triangular or double-peak shape persists over a wide range of angles (1.6°–3.1°) without inverting ("downward triangle"), and the calibration method in Fig. 3 is effective for stoichiometry control between ~2–2.8°. In contrast, we find (110) alignment has a narrower useful range, consistent with ref. 48.

In conclusion, motivated by the synchrotron surface diffraction approach, we exploit RHEED signatures of intrinsic layer inversion during shuttered growth to control stoichiometry and termination for a number of perovskite phases. By analyzing diffracted-beam rocking curves at different mixed terminations, we develop a phenomenological model that explains intensity oscillations at a fixed incidence angle, enabling precise determination of shutter times for relative (A:B = 1) calibration; absolute flux calibration is achieved by flux-matching and measuring the period of codeposition oscillations. The successful growth of high-quality, stoichiometric films across multiple perovskite phases suggests this method could be extended to the shuttered growth of other layered materials in which layer inversion occurs.

## Methods
### Film growth
The MBE films presented here have been grown in three different systems: (1) a GenExplor MBE system (Veeco, Inc.) at UBC/QMI (Vancouver) using molecular oxygen or an RF oxygen plasma; RHEED was performed at 10 keV (Staib Instruments, Inc.) and data was collected with a 14-bit CCD and analyzed by kSA 400 software in real time (kSpace Associates, Inc.)[64]; (2) an in-house designed oxide MBE system at the CNR-IOM (BEAR and APE beamlines of Elettra Synchrotron, Trieste) using molecular oxygen or distilled pure ozone[49,65]; and (3) a DCA oxide MBE system at MPI-FKF (Stuttgart) using distilled pure ozone[73,74]. Substrate temperatures and oxygen partial pressures in Figs. 1 and 2 are $P_{O2} = 5 \times 10^{-6}$ Torr and 780 °C (measured by a thermocouple near the substrate, GenXplor) or 730 °C (measured by an optical pyrometer, Trieste); $P_{O2} = 5 \times 10^{-6}$ Torr and 680 °C in Fig. 3;

$P_{O2} = 4 \times 10^{-6}$ mBar, $T_{substrate} = 780$ °C in Fig. 4; and elsewhere as indicated. PLD films shown in the Supplementary Fig. 12 have been grown in a multitarget Neocera system at Temple University with excimer wavelength 248 nm (KrF) and differentially-pumped RHEED.

### X-ray diffraction
XRD characterization was performed in Trieste using a Philips Xpert with Cu $K_{\alpha1}$ radiation (wavelength 1.5406 Å) and high-resolution optics with incident 4-bounce Ge(220) symmetric and diffracted 2 bounce Ge(220) asymmetric monocromators, or only incident monocromator for reciprocal space maps, and point-proportional detector; and a Bruker D8 Discover at UBC/QMI with Cu $K_{\alpha1}$ radiation, high-resolution source optics (focusing mirror and 2-bounce Ge(220) monochromator) and a Detrix 2D detector.

### Atomic force microscopy
AFM measurements were performed in ambient conditions in semi-contact mode on a Solver pro instrument (NT-MDT) using commercial silicon cantilevers (NSG30 NT-MDT, radius of curvature <10 nm, spring constant = 40 N/m).

### Rutherford backscattering spectrometry
RBS was performed in backscattering geometry with 2.275 MeV He$^+$ ions at Legnaro National Lab, Padova, Italy, and 2.0 MeV He$^+$ at the Department of Physics, Rutgers University, Piscataway, NJ.

### Scanning transmission electron microscopy
For the electron-transparent TEM sample, a standard sample preparation procedure including mechanical grinding, tripod wedge polishing and argon ion milling with a liquid nitrogen cooled stage was performed. A precision low-temperature ion polishing system (PIPS II, Model 695) was used for argon ion thinning. High-resolution STEM analyses were performed with a JEOL JEM-ARM200F equipped with a cold field emission electron source and a probe $C_s$ corrector (DCOR, CEOS GmbH) at an acceleration voltage of 200 kV. STEM-HAADF and EELS acquisition was performed with convergence semi-angles of 20 and 28 mrad, resulting in probe sizes of ~0.8 Å and ~1.0 Å, respectively. The collection angles for the HAADF images ranged from 75 to 310 mrad, while a collection semi-angle of 111 mrad was used for the EELS investigations. For the elemental maps, EELS spectrum images (SIs) were acquired with a dispersion of 0.5 eV/channel, in the 250-2105 eV energy range containing the elemental ionization edges of interest. Sr, Ti and O maps were constructed from the integrated inelastic intensities of the background-subtracted Sr $L_{3,2}$ (1930-2045 eV), Ti $L_{3,2}$ (448-479 eV) and O $K$ (520-560 eV)-edges, respectively. SIs were processed with Principal Component Analysis (PCA) using the temDM Multivariate Statistical Analysis plugin for DigitalMicrograph (available at temdm.com) in order to reduce their noise content.

## Data availability
Source data are provided with this paper (ref. 75).

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

## Acknowledgements

This research was funded in part by the Max Planck-UBC-UTokyo Centre for Quantum Materials and the Canada First Research Excellence Fund, Quantum Materials and Future Technologies Program. The work at UBC was also supported by the Natural Sciences and Engineering Research Council of Canada (NSERC) and the Canada Foundation for Innovation (CFI). B.A.D. was funded in part by a QuantEmX grant from ICAM for growths at MPI-FKF (Stuttgart, Germany) and the Gordon and Betty Moore Foundation through Grant GBMF5305. RBS measurements were performed (1) at the INFN-Legnaro National Laboratories (Padova, Italy) under the proposal "OxideRBS", for which B.A.D. gratefully acknowledges assistance from Dott.ssa Marina Berti (U. Padova); and (2) by Ryan Thorpe (Rutgers U.) in the Experimental Surface Science Group. K. Wijesekara, D. Yang, Q. Lei and M. Golalikhani of Temple University are acknowledged for their assistance during PLD film growth. B.A.D. and A.Y.P. acknowledge support from the FVG Regional Project SPINOX funded by Legge Regionale 26/2005 and the decreto 2007/LAVFOR/1461.

## Author contributions

B.A.D. and A.Y.P. conceived the project, and B.A.D., A.Y.P., E.B. and K.Z. supervised the design of the experiments. B.A.D., A.Y.P., Y.E.S. and K.Z. wrote the manuscript. B.A.D., A.Y.P., F.L., R.P., H.S. and C.L. grew the films and performed the RHEED analysis. A.Y.P. and B.A.D. performed the XRD characterization and analysis. P.P. performed the AFM characterization and analysis. P.S.-L., Y.E.S. and P.A.v.A. performed the STEM characterization and analysis. All authors, including G.C., G.L., G.K., X.X. and P.T., discussed the experiments and results, and corrected the manuscript.

## Funding

## Competing interests

The authors declare no competing interests.
