## [Transparent Peer Review file · Nature Communications]

A universal method for in situ control of stoichiometry and termination of epitaxial perovskite films

Corresponding Author: Dr Bruce Davidson

Version 0:

Reviewer comments:

Reviewer #1

(Remarks to the Author)

See attached file

Reviewer #2

(Remarks to the Author)

Dear Editor,

This study investigates “a universal method for precise control of perovskite films”, which is very interesting. It is claimed that “...providing for the first time a completely in situ method...”. However, the main conclusion cannot be fully supported by the present data. Therefore, before recommending its publication, some important concerns need to be addressed.

(1) The “universal method” are key word in this work. Could the authors clearly define what the universal method is? It is very hard to follow this concept in the present manuscript.

(2) In Fig. 1 (c), why does the RHEED intensity (red) first increase and then decrease? Does this behavior strongly depend on the incident angle?

(3) Generally, the oscillation of RHEED intensity is closely connected to the surface roughness of films. Therefore, the model in Fig. 2 (a) is not common. Is there any theory or calculation to support this model?

(4) As the authors mentioned, the lineshape of the RHEED curve is very complex and sensitive to several parameters. Therefore, it is NOT sufficient to show RHEED data only in the manuscript. To demonstrate the precise control of perovskite films, I strongly suggest the authors add comprehensive measurements of scanning transmission electron microscope (STEM) and atomic force microscope (AFM).

Reviewer #3

(Remarks to the Author)

Dear Editor, the manuscript by Davidson et al carefully describes a method to use the evolution of RHEED diffraction spots to precisely monitor the deposition rates of perovskite oxides during shuttered MBE growth. The text and accompanying figures give a full account of the method including validation in terms of crystal structure and stoichiometry. As such, the manuscript will be an invaluable guide for MBE users who want to adopt this method. However, I do not think that in its current form, the manuscript is suitable for Nat. Comm. given its specialist style as well as the lack of validation in terms of the measurement of functional properties. Therefore I recommend to publish in a more specialized journal. One additional minor comment: upon re-submission elsewhere, please take care of the authors notes in the text (in red) of the Supporting information.

Version 1:

Reviewer comments:

Reviewer #1

(Remarks to the Author)

The authors have addressed all my concerns. At its current form, the manuscript shows noteworthy results which will be of great significance to control and understand the growth of oxide thin films.

Minor comment: Fig 3: it is difficult to distinguish the color for "Shutters closed" from "Ti shutter open". According to my view, after addressing this "typo" the paper can be accepted for publication.

Reviewer #2

(Remarks to the Author)

Dear Editor,

The authors have responded to most comments. However, it is strongly recommended that they provide a concise definition of the "universal method" concept and enhance the clarity of all figure captions. The current version of the manuscript is hard to follow. The plotting of figures can be further improved. Importantly, due to all five figures being based on the analysis of RHEED, the setup and measurements of RHEED should be added in detail. For example, how to calibrate and measure the incident angles of RHEED?

Reviewer #3

(Remarks to the Author)

Dear Editor, having reviewed the manuscript and the responses to the critics of the reviewers in the previous round I conclude the following:

1) I agree with referee #3 that although the work is has been performed and described meticulously, the work is more suitable for a specialist journal. The group of growers that uses these kinds of methods is relatively small and would benefit from this kind of methodological and systematic description. For the more general scientific community it would be hard to assess the benefits of the described method. Also, it not clear whether methodology is at all suitable for industrial scale up, rendering the target audience to mostly specialist scientist.

2) The authors respond well to the referees detailed and content-related comments

3) One small remark regarding the evidence of 'crystal quality' brought up by the authors: XRR merely probes differences in (optical) density and therefor the absence of Kiessig fringes is not a measure of crystal quality.

Version 2:

Reviewer comments:

Reviewer #2

(Remarks to the Author)

This manuscript can benefit the MBE community.

REVIEWER COMMENTS

Reviewer #1 (Remarks to the Author):

The authors develop a beautiful and precise approach to control and in-situ monitoring the deposition of perovskites during shuttered growth. SrTiO₃ is explored and presented in detail but it is applied to many other perovskites. So, the authors have done a meticulous work and the methodology seems universal for perovskites.

We thank Reviewer #1 for confirming the scope and importance of the results in the manuscript.

On the other hand I think there are many useful and relevant information in the supplemental information that should be included in the main manuscript. For example, a combination of figure S3 and S7 could help understanding the impact of the shutter method on crystallinity and surface morphology.

Also, I consider mandatory to include high resolution TEM images from a titanate film grown on SrTiO₃ with TiO₂ and mixed termination.

It will be a good add to include TEM images of a samples grown with the sequence: triangular, start double-peak, shallow double peak and deep double peak stages represented in figure 2. This will facilitate the correlation of the crystalline order observed from RHEED and TEM. How difficult would be to stop the growth after the triangular sequence and perform TEM? Would the structure be stable?

At the risk of lengthening the manuscript, we have added a new figure (Fig. 4 on pg. 10) that combines new TEM and EELS results with the AFM results previously reported in Supplemental Figure 7. We have also added text to integrate these new results into the revised manuscript (lines 234-242 and 246-249). The TEM HAADF and EELS analysis confirms the excellent epitaxial quality and stoichiometry of films grown by the shutter method, using different terminations and different double-peak shapes: first, 25uc using Sr₁/Ti₁ cycles starting from a mixed termination (~0.5 Sr layer on TiO₂, and “shallow double-peak” oscillations); and second, 25uc using Sr_{2.5}/Ti_{2.5} cycles on TiO₂ termination (“deep double-peak” oscillations). In the TEM HAADF images, both layers show defect-free and an ideal layered AO–BO₂ 113 structure; indeed, the layers are indistinguishable. EELS analysis shows uniform Sr and Ti throughout the film (including the interface between the first Sr₁/Ti₁ and second Sr_{2.5}/Ti_{2.5} layers), with the exception of a small dip in Ti and O intensities at the film/substrate interface from disorder at the starting substrate surface. This supports the conclusions drawn from the RHEED analysis in the manuscript based on layer inversion during growth cycles.

It can be seen that the shutter method produces layers of equal quality to the substrate bulk crystal, and suggests this method can be used to grow buffer layers of superior quality and controllable surface termination for subsequent film growth (as done routinely in semiconductor heterostructures by MBE), offering an alternative to growth directly on the substrate surface whose properties are often not well known.

We note that the AFM results (new Fig. 4, old Fig. S7) show the STO film morphology after stopping growth at different points in the cycle; the step-and-terrace structure of the starting substrate is seen also in the SrO-terminated surface up to ~1.3 layers SrO/TiO₂. This is consistent with the known stability of the STO surface in air for AFM and TEM measurements.

Minor: revise Figure S4 (e) and S5(c) captions “Sasha-growth conditions” as there are no such panels in the figure.

This text has been removed.

Reviewer #2 (Remarks to the Author):

Dear Editor,

This study investigates “a universal method for precise control of perovskite films”, which is very interesting. It is claimed that “...providing for the first time a completely in situ method...”. However, the main conclusion cannot be fully supported by the present data. Therefore, before recommending its publication, some important concerns need to be addressed.

(1) The “universal method” are key word in this work. Could the authors clearly define what the universal method is? It is very hard to follow this concept in the present manuscript.

We appreciate Reviewer #2's comment here, and we have taken this opportunity to slightly reword the corresponding manuscript sections “Results (i), (ii) and (iii)” in order to clarify the description of the “universal method”. (The rewording is sufficiently pervasive but subtle that the changes have not been highlighted) The manuscript covers a lot of territory to explain how layer-inversion is seen in the RHEED rocking curves, and how these signatures provide for real-time feedback to identify the stoichiometric growth of shuttered unit cells. The rocking curves in Figure 1 are used to identify layer-inversion and to select diffraction conditions giving the largest intensity oscillations during shuttered cycles. Based on the systematic change in shape of these oscillations for different x in Sr_x/Ti_x starting from different mixed terminations, Figure 2 describes the model that is developed into a universal calibration method described in Figure 3. The universal method is described in more detail in Supplementary Note 1.

In brief, the “universal method” is presented using the example of SrTiO_3 that is doped with La and other rare-earth elements on the A-site to stabilize the entire phase diagram e.g. $\text{Sr}_{1-x}\text{La}_x\text{TiO}_3$ for any x ($0 \leq x \leq 1$):

- 1) the relative calibration of Sr to Ti fluxes is achieved using a stable “double-peak” growth regime, i.e. depositing Sr_x/Ti_x cycles ($x \sim 1$) starting from a mixed termination (~ 0.5 SrO layers on TiO_2 termination) and regulating the shutter times to maintain a constant oscillation shape;
- 2) the absolute calibration of Sr and Ti fluxes is achieved by first matching Sr and Ti shutter times and hence fluxes (by adjusting cell temperatures) and then measuring the oscillation period during codeposition;
- 3) the La flux is calibrated by partial substitution of La for Sr, employing the relative calibration of step (1), opening Sr and La shutters simultaneously during the A- part of the A_x/B_x cycle.

We hope these rewritten sections of the manuscript provide a clearer description of the method.

(2) In Fig. 1 (c), why does the RHEED intensity (red) first increase and then decrease? Does this behavior strongly depend on the incident angle?

In the Discussion, have added new text (lines 288-299) that explains in more detail the behavior of the RHEED intensity oscillations during shuttered cycles, as described below.

As shown in Supplementary Figure 9, the behavior of the diffracted intensity during Sr deposition depends strongly on angle. At the diffraction geometry used in Figures 2 and 3 (i.e., incidence angle $\sim 2.5^\circ$ and (100) azimuth), the diffracted intensity increases strongly until ~ 1.3 layers of Sr are deposited (the “inflection point”) and then decreases for additional Sr deposition. The increasing intensity is explained as a combination of scattering factor and RHEED surface resonance condition (see Discussion); the subsequent decreasing intensity is explained by increasing surface roughness as the second SrO rock-salt layer forms (see the AFM images in Figure 4e). Combining this behavior with layer-inversion during Ti deposition gives the explanation for the “double-peak” shape used in

the model (Figure 2) and calibration method (Figure 3). At different angles, the diffracted intensity can increase more slowly or even decrease during Sr deposition, and the inflection point can be entirely missing.

As the new paragraph in the Discussion section describes, evidence for a surface resonance at the chosen incidence angle is supported by the experimental observation that the double-peak shape persists as the azimuthal angle is rotated toward the “one-beam condition”. (See also our reply to the next comment on calculations to support the model of Figure 2).

(3) Generally, the oscillation of RHEED intensity is closely connected to the surface roughness of films. Therefore, the model in Fig. 2 (a) is not common. Is there any theory or calculation to support this model?

We first note that there are no calculations in the literature to explain the model in Figure 2 or the data in the manuscript. The referee is correct that the *specular* RHEED intensity oscillations are usually explained as resulting from scattering from step-edges, whose density varies during layer-by-layer growth and can be correlated to surface roughness. Experimentally, the *diffracted* beam intensity is not nearly as sensitive as surface roughness as the specular intensity (we give references in the manuscript). In fact, the peculiar diffracted intensity oscillations seen in Figures 2 and 3 have never been seen or reported in the literature for specular intensities. As we mention in the manuscript (lines 51-54), dynamical scattering calculations attempting to replicate the diffracted intensity oscillations seen in Figures 2 and 3 would be extremely complicated to perform, as they would require knowledge of the positions of all atoms in the first several layers for all partial coverages during the shuttered cycle. In fact, only in the “one-beam” condition is this constraint (approximately) relaxed, and we have begun to explore calculations in this geometry to explain our data for double-peak oscillations in this condition. These calculations by themselves are sufficiently complicated that we consider them to be outside the scope of this manuscript, and will be the subject of a future publication

(4) As the authors mentioned, the lineshape of the RHEED curve is very complex and sensitive to several parameters. Therefore, it is NOT sufficient to show RHEED data only in the manuscript. To demonstrate the precise control of perovskite films, I strongly suggest the authors add comprehensive measurements of scanning transmission electron microscope (STEM) and atomic force microscope (AFM).

Please see our response to Referee #1 regarding the new TEM results that we have added as a new figure (Fig. 4) with accompanying text (lines 241-249 and 253-256). In addition, we have moved the AFM results from Supplementary Figure 7 into the main text and show these together with the TEM results.

Reviewer #3 (Remarks to the Author):

Dear Editor, the manuscript by Davidson et al carefully describes a method to use the evolution of RHEED diffraction spots to precisely monitor the deposition rates of perovskite oxides during shuttered MBE growth. The text and accompanying figures give a full account of the method including validation in terms of crystal structure and stoichiometry. As such, the manuscript will be an invaluable guide for MBE users who want to adopt this method. However, I do not think that in its current form, the manuscript is suitable for Nat. Comm. given its specialist style as well as the lack of validation in terms of the measurement of functional properties. Therefore I recommend to publish in a more specialized journal. One additional minor comment: upon re-submission elsewhere, please take care of the authors notes in the text (in red) of the Supporting information.

We appreciate Reviewer #3's comments here, and have made several modifications to the manuscript to accommodate them. First, we disagree that our new shuttered method applies only to the MBE technique; we demonstrate that it applies also to PLD (Supplementary Figure 8) using sequential ablation from binary oxide targets. We have emphasized this point in the revised text with additional text (lines 267-270). In so far as the layer inversion mechanism is universal, we expect the shutter method can be applied to other deposition techniques as well e.g. sputtering, CVD, ALD etc. to improve stoichiometry control, and not only MBE as Reviewer #3 suggests. This broadens the scope of our results and, we argue, makes them suitable for publication in Nature Communications.

In response to Reviewer #3's request for measurement of the functional qualities of films produced by this method, we have included new text with references to state-of-the-art functional properties of magnetic tunnel junctions grown by the shutter method (lines 260-262) and highlighting the interest in the field of high-entropy oxides with compositions identical to what we have grown (lines 264-265).

REVIEWER COMMENTS for “A universal method for in situ control of stoichiometry and termination of epitaxial perovskite films”, B. A. Davidson et al, Nature Communications, May 29th, 2025

Reviewer #1 (Remarks to the Author):

The authors have addressed all my concerns. At its current form, the manuscript shows noteworthy results which will be of great significance to control and understand the growth of oxide thin films. Minor comment: Fig 3: it is difficult to distinguish the color for "Shutters closed" from "Ti shutter open". According to my view, after addressing this "typo" the paper can be accepted for publication.

The authors thank the reviewer, and we have fixed the “typo” in Figure 3.

Reviewer #2 (Remarks to the Author):

Dear Editor,

The authors have responded to most comments. However, **(1)** it is strongly recommended that they provide a concise definition of the "universal method" concept and **(2)** enhance the clarity of all figure captions. The current version of the manuscript is hard to follow. **(3)** The plotting of figures can be further improved. **(4)** Importantly, due to all five figures being based on the analysis of RHEED, the setup and measurements of RHEED should be added in detail. For example, how to calibrate and measure the incident angles of RHEED?

We appreciate the reviewer’s feedback and agree that **(1)** a concise definition of the “universal method” concept would improve the clarity of the manuscript. We have now added a short definition (lines 191-197) in the text introducing Figure 3, to better frame the concept for the reader. Regarding the figure captions, we recognize they are dense and detailed. However, with regard to comment **(2)**, many critical experimental details are currently included in the captions rather than the main text, which helps keep the manuscript focused on the central storyline. Simply cutting down the captions may risk losing important information or force a substantial restructuring of the main text, which could make the manuscript more difficult to follow overall. That said, we have carefully reviewed the captions and made minor edits to improve readability without sacrificing essential information. In addition, with regard to comment **(3)**, we have cleaned up several details in the figures (e.g. units on the x-axis of Fig. 5, color legend for Fig. 3, region of interest for integrating the RHEED intensity) that should improve readability.

In response to comment **(4)** about the RHEED methods, we have now added a description of the RHEED setup and measurements in a new Supplementary Figure 1, including calibration of the incident angle, and the definition of the regions of interest monitored in the RHEED image during growth. We believe this addition will help readers unfamiliar with RHEED better understand the figures and analysis in the main text. We would also like to clarify that while the reviewer asks about calibration of the incidence angle, the precise calibration is not critical for our method. Using the diffraction conditions specified in the main text, in which the specular spot is set to the intersection with the Kikuchi lines, is sufficient to ensure the appropriate geometry for performing the universal calibration method.

Reviewer #3 (Remarks to the Author):

Dear Editor, having reviewed the manuscript and the responses to the critics of the reviewers in the previous round I conclude the following:

1) I agree with referee #3 that although the work is has been performed and described meticulously, the work is more suitable for a specialist journal. The group of growers that uses these kinds of methods is relatively small and would benefit from this kind of methodological and systematic description. For the more general scientific community it would be hard to assess the benefits of the described method. Also, it not clear whether methodology is at all suitable for industrial scale up, rendering the target audience to mostly specialist scientist.

2) The authors respond well to the referees detailed and content-related comments

3) One small remark regarding the evidence of 'crystal quality' brought up by the authors: XRR merely probes differences in (optical) density and therefor the absence of Kiessig fringes is not a measure of crystal quality.

We respectfully disagree with the assessment that our method is limited to a narrow specialist audience, as stated in comment (1). Although our work is rooted in MBE methodology, we demonstrate its broader applicability by successfully extending the shuttered growth concept to pulsed laser deposition (PLD) using sequential ablation from binary oxide targets (main text lines 253-256, and Supplementary Figure 9).

Furthermore, because the underlying layer inversion mechanism we identified is universal for the perovskite phases studied, the principles of our shuttered method are expected to apply to other deposition techniques — including sputtering, hybrid MBE, chemical vapor deposition (CVD), atomic layer deposition (ALD), and potentially others — where stoichiometry control is critical. Thus, the broader significance of our findings goes beyond the MBE community alone.

With regard to questions of industrial scale-up, there is a discussion in the final paragraph of Supplementary Note 1 that addresses exactly the question of adapting the calibration method to large wafers for which sample rotation is essential to achieve uniform deposition of layers.

We agree with the reviewer's remark about XRR measurements. By combining RHEED, XRR and XRD measurements, we believe the evidence provided sufficiently supports the conclusions drawn regarding crystal quality.

We believe this expanded relevance makes our work suitable for a wide readership and appropriate for publication in Nature Communications.

The authors develop a beautiful and precise approach to control and in-situ monitoring the deposition of perovskites during shuttered growth. SrTiO₃ is explored and presented in detail but it is applied to many other perovskites. So, the authors have done a meticulous work and the methodology seems universal for perovskites.

On the other hand I think there are many useful and relevant information in the supplemental information that should be included in the main manuscript. For example, a combination of figure S3 and S7 could help understanding the impact of the shutter method on crystallinity and surface morphology.

Also, I consider mandatory to include high resolution TEM images from a titanate film grown on SrTiO₃ with TiO₂ and mixed termination.

It will be a good add to include TEM images of a samples grown with the sequence: triangular, start double-peak, shallow double peak and deep double peak stages represented in figure 2. This will facilitate the correlation of the crystalline order observed from RHEED and TEM. How difficult would be to stop the growth after the triangular sequence and perform TEM? Would the structure be stable?

Minor: revise Figure S4 (e) and S5(c) captions “Sasha-growth conditions” as there are no such panels in the figure.